# Modelling Neurological Diseases in Large Animals: Criteria for Model Selection and Clinical Assessment

**DOI:** 10.3390/cells11172641

**Published:** 2022-08-25

**Authors:** Samantha L. Eaton, Fraser Murdoch, Nina M. Rzechorzek, Gerard Thompson, Claudia Hartley, Benjamin Thomas Blacklock, Chris Proudfoot, Simon G. Lillico, Peter Tennant, Adrian Ritchie, James Nixon, Paul M. Brennan, Stefano Guido, Nadia L. Mitchell, David N. Palmer, C. Bruce A. Whitelaw, Jonathan D. Cooper, Thomas M. Wishart

**Affiliations:** 1Royal (Dick) School of Veterinary Studies and Roslin Institute, University of Edinburgh, Easter Bush Campus, Roslin, Midlothian EH25 9RG, UK; 2Medical Research Council Laboratory of Molecular Biology, Francis Crick Avenue, Cambridge CB2 0QH, UK; 3Centre for Clinical Brain Sciences, University of Edinburgh, Chancellor’s Building, 49 Little France Crescent, Edinburgh EH16 4SB, UK; 4Department of Clinical Neurosciences, NHS Lothian, 50 Little France Crescent, Edinburgh EH16 4TJ, UK; 5The Large Animal Research & Imaging Facility, Royal (Dick) School of Veterinary Studies and Roslin Institute, University of Edinburgh, Easter Bush Campus, Roslin, Midlothian EH25 9RG, UK; 6Translational Neurosurgery, Centre for Clinical Brain Sciences, University of Edinburgh, Edinburgh EH16 4SB, UK; 7Bioresearch & Veterinary Services, University of Edinburgh, Chancellor’s Building, 49 Little France Crescent, Edinburgh EH16 4SB, UK; 8Faculty of Agriculture and Life Sciences, Lincoln University, P.O. Box 85084, Lincoln 7647, New Zealand; 9Departments of Pediatrics, Genetics, and Neurology, Washington University School of Medicine in St. Louis, 660 S Euclid Ave, St. Louis, MO 63110, USA

**Keywords:** neurological disease, large animal model, clinical assessment, model selection criteria

## Abstract

*Issue:* The impact of neurological disorders is recognised globally, with one in six people affected in their lifetime and few treatments to slow or halt disease progression. This is due in part to the increasing ageing population, and is confounded by the high failure rate of translation from rodent-derived therapeutics to clinically effective human neurological interventions. Improved translation is demonstrated using higher order mammals with more complex/comparable neuroanatomy. These animals effectually span this translational disparity and increase confidence in factors including routes of administration/dosing and ability to scale, such that potential therapeutics will have successful outcomes when moving to patients. Coupled with advancements in genetic engineering to produce genetically tailored models, livestock are increasingly being used to bridge this translational gap. *Approach:* In order to aid in standardising characterisation of such models, we provide comprehensive neurological assessment protocols designed to inform on neuroanatomical dysfunction and/or lesion(s) for large animal species. We also describe the applicability of these exams in different large animals to help provide a better understanding of the practicalities of cross species neurological disease modelling. *Recommendation:* We would encourage the use of these assessments as a reference framework to help standardise neurological clinical scoring of large animal models.

## 1. Introduction

Neurological disorders affect one in six people globally and are the leading cause of disability-adjusted life-years (DALY) that accounts for the combined burden of disease, health risks, and premature death [1,2,3]. This equates to nearly 1 billion people worldwide with conditions caused by a broad range of factors which ultimately result in the altered structure and/or function of the nervous system. These conditions can vary greatly in terms of aetiology, age of onset, and duration, but collectively represent a significant healthcare burden. The United Kingdom’s publicly funded National Health Service spends approximately GBP 4.4 billion annually on the treatment and management of neurological conditions, the vast majority of which are currently incurable [1]. Brain disorders in the United Kingdom alone are estimated to exceed a cost of GBP 100 billion per year [4]. Effective therapeutic strategies are therefore in high and urgent demand.

In the field of neurodegenerative research, translation of potential therapeutics from the laboratory to the clinic has a high failure rate. One reason for this failure is an over-reliance on rodent model systems, which often do not recapitulate key aspects of the human condition of interest, and are developed in the physiological context of a small, nocturnal prey species [5,6,7]. Such work is critical in advancing our understanding of the molecular mechanisms governing the function and dysfunction of the mammalian nervous system. However, it is increasingly accepted that the use of larger, more physiologically complex model systems is required to effectively assess the translational relevance of novel therapeutic developments for human interventions. Using more physiologically relevant model systems can reduce waste in translational research and increase the likelihood of promising treatments achieving regulatory approval [8,9].

Whilst some might consider the choice of one animal model system over another to be equally unpalatable, it matters when this has the potential to efficiently generate more clinically relevant data, with a smaller number of animals used in fewer, more targeted experiments. It also matters when the same outputs can be used to enhance life quality and advance treatment for veterinary patients with similar neurological disorders. A ‘better’ model, coupled with careful study design and species–specific ethical considerations guided by specialists trained in comparative neuroanatomy, behaviour, physiology, and neuropathology has the power to truly embrace 3Rs objectives [10]. ‘Larger’ does not necessarily mean ‘better’, and ‘better’ requires evidence and validation.

Any model system is limited not only by its physiological representation of the condition in question, but also by our understanding of that disease. From the outset, a detailed and robust characterisation is necessary to determine how effectively the model reproduces the pathophysiological hallmarks of the corresponding human condition, but also to define ‘biomarkers’ for therapeutic intervention. Biomarkers of disease could include behavioural, imaging-based, or molecular markers, or (ideally) a combination of these features, that should track with disease progression and provide an insight to the ideal time when therapies should be administered to maximise their efficacy. Sometimes, the distinct consequences of a human mutation in a non-human animal can be as insightful for understanding the human disease as any similarities—it is therefore vital that any observations diverging from those expected are reported with as much rigour and care as the similarities, and not simply discarded as an aspect of ‘model failure’.

Given the complexity and composition of the nervous system, behavioural or neurological tests are often the first line of assessment in diagnosing neurological disorders in human patients [11]. Assessment of species–species behaviour and mental status also forms an important and early aspect of managing patients in the veterinary clinical setting [12]. Followed by a complete general physical and systematic neurological evaluation (some aspects of which must be tailored to the species), these examinations can confirm whether a neurological deficit exists, and provide an accurate neuroanatomical diagnosis or limit the possible causes of clinical signs [13]. Repeating these assessments over time can monitor how a neurological disease progresses and importantly, the effectiveness of any therapeutic intervention [14].

Recent developments in genome editing have given us the ability to generate bespoke mutations in almost any species required in order to fill translational gaps with physiologically relevant models [15]. The most ethically viable and physiologically appropriate systems for neurological modelling are of livestock origin (sheep and pig) [16,17,18,19]. Yet, there are a limited number of publications that cover neurological assessment specifically in livestock [13,14,20,21,22,23]. This reflects the reality that most livestock presenting with neurological disease would usually be culled prior to or as a result of in-depth examinations, since treatment of these cases is rarely justifiable on welfare, biosecurity, and/or financial grounds [24]. This in turn leads to fewer veterinary neurologists who specialise in production animal neurology, since the demand for this expertise is substantially lower than in companion animal medicine.

Here, we discuss the key factors relevant to model selection with appropriate options for premortem assessments that would enable comprehensive and longitudinal monitoring of a neurological phenotype in large animal models. Species–specific considerations (such as size and demeanour) with implications for their implementation will also be highlighted, with a focus on production animals. Postmortem extraction, processing, preservation, and neuropathological analysis of tissues are critical aspects to optimise when validating any disease model, but these are covered well in other texts [25,26,27] and are beyond the scope of this work. Likewise, although several human neurological conditions have their counterparts in equids [13,28], for financial and safety reasons, these species are not a practical choice for comparative study beyond the veterinary clinical research setting and are therefore not covered in any detail here.

## 2. Model Selection

***What is the need?*** Research underpinning therapeutic development, particularly in the field of neurodegenerative conditions, requires animal models. Some fundamental research questions can be answered using culture-based systems, but the complexity of the human nervous system, coupled with the (now accepted) multi-systemic nature of many of these conditions, means that appropriate and effective therapeutic development mandates in vivo systems [29,30,31,32]. This includes, but is not limited to, considerations such as scaling up, routes of administration (especially important with viral-mediated therapeutics), distribution and persistence, immune response, and physiological coupling to the day–night cycle [33].

***What are the options?*** In order to address research needs for clinical development there are a few options in which large animals can be used where appropriate:

***Using diseases of large animals to inform on human disease***—when assessing naturally occurring neurological conditions in domestic species, veterinarians typically classify neurological diagnoses according to the ‘VITAMIN D’ acronym (Table 1). For each of these categories, there are notable examples of translational relevance. There are obvious gains to be made from studying these clinical cases as they arise, and in considering the potential for human and veterinary medical developments based on repurposing of successful animal intervention trials. However, the sporadic nature of neurological case presentation and the inherent challenges of veterinary clinical research (with ethical considerations and breed-related factors) reduce the opportunities to study clinical populations with adequate control and statistical power to meaningly inform human medicine and clinical practice on a desirable timescale.

***Spontaneous mutations in animal populations***—animal models of neurological disease that exist naturally are an obvious study choice, especially for those conditions where there are multiple genetic and environmental factors which impact on disease phenotype. These models have worked successfully in certain conditions where scientists have been notified of neurological phenotypes and, following appropriate genetic testing, a specific breeding cohort has been established for future studies (this is true of canine narcolepsy, muscular dystrophy in Golden Retriever dogs; and CLN5 and CLN6 neuronal ceroid lipofuscinosis (NCL) in South Hampshire and Merino sheep, respectively [81,82,83]). However, due to the low incidence of animals which harbour mutations causing these conditions naturally, their identification and ability to generate a breeding nucleus is problematic. This is compounded by late presentation and/or diagnosis by which point breeding becomes unethical, or the animals have already been neutered such as in CLN1 NCL disease in canines [84,85].

Sporadic cases can be developed as a research tool. For example, a promising gene-editing strategy to restore dystrophin expression was developed in dogs with a canine version of Duchenne muscular dystrophy [86]. Similarly, enzyme replacement therapy trials in the naturally occurring CLN2-deficient Dachshund model of NCL directly led to successful implementation of cerliponase alfa (Brineura) treatment in patients with early results indicating a cessation or slowing of disease progression [87,88,89]. Trials in these animals has informed on clinical trial design, dosage, and routes of administration in patients producing successful outcomes [88]. This led to Brineura obtaining Food and Drug Administration (FDA) approval in a significantly shorter timeframe [88]. In another example, studies in sheep with endemic scrapie have determined that blood is infectious in animals devoid of clinical symptoms, which has fundamentally changed blood donation protocols and legislation in light of the Variant Creutzfeldt–Jakob disease (vCJD) epidemic [89,90]. Moving forward, collaborations are combining expertise in human and veterinary neurology to elucidate molecular mechanisms of immune-mediated encephalitis for the benefit of both human and feline patients [91,92,93].

However, given the limitations around the sporadic nature of these useful mutations appearing and the fact that some therapeutic interventions require the exact mutation as identified in the patient population, it is often necessary to generate bespoke models. This approach is beneficial for identification of any altered molecular pathway(s) and/or cellular functions that are perturbed in vivo yet are not present in transgenic knockout or spontaneously arising animals that are phenotypically similar to its human counterpart, but with a distinct genetic aetiology [94].

***Bespoke model systems***—Tailor-made models of human disease are now well established [15,95]. Genetic engineering of livestock places specific disease-causing gene into species that can more effectively model human physiology than rodents [17,18,19]. However, the generation of a new model for any neurodegenerative condition is a decision which should not be taken lightly.

Benefits of generating models of neurological disorders in large animals have already been reported [8]. A new model of ataxia telangiectasia in pigs has given novel insights into cerebellar Purkinje cell loss at birth, which was not observed in the murine model, as well as producing a multi-systemic phenotype as documented in human disease [19,96]. An ovine model of CLN1 disease produced a lifespan, clinical presentation, and duration that more closely resembles the human disease. Cerebellar density is altered similarly to humans; a feature not observed in the rodent models [17]. A porcine model of neurofibromatosis type 1 mimics more of the hallmarks of human disease than the heterozygous mouse models including the classical café au lait macules and neurofibroma phenotypes [18]. Such improvements in clinical phenotypes observed in large animal models can not only provide a better understanding of disease but importantly an opportunity for increased translational therapeutic efficacy on this intermediary platform [9]. There are also practical and translational advantages that result from using large animal models; due to their more complex brain and body size they are better suited to testing drug delivery systems and carry out therapeutic biodistribution studies [97].


**
*Considerations:*
**


***Ethical***—must be a priority when modelling any disease with either ‘spontaneous’/naturally occurring systems or bespoke engineered animals (for which, a critical need that cannot otherwise be satisfied must exist). This is especially important for neurological disorders which often have a devastating phenotype and/or result in neurological deficits that impact life quality long before a humane endpoint is reached. The potential clinical benefit needs to be weighed against the lived experience cost to the individual animal; the combined expertise of experienced animal handlers, veterinarians with specialist skills both in neurology and the species of interest, and personnel trained in animal welfare and ethics are essential. Appropriate experimental end points must be established during the study design and are typically much earlier than would necessitate euthanasia in the veterinary clinical setting. Ultimately, clinical success in neurological patients will be best achieved by early disease detection and early intervention. As our ability to identify and treat the earliest stages of disease in appropriate animal models increases, the need and justification for progressing research animals to a humane end point will diminish.

***Legislation and guidelines***—Appropriate legislative and welfare measures must be in place when defining humane end points and this will differ from country to country. In the United Kingdom, for example, horses, dogs, and cats (in addition to non-human primates) are specially protected and subject to even stricter regulations that other species. The establishment of a National Centre for the Replacement Refinement and Reduction of Animals in Research [10] further encourages best practice through dissemination of the ARRIVE and PREPARE Guidelines [10,98,99]. In the UK, the use of animals for scientific purposes is regulated by the Animals (Scientific Procedures) Act 1986; however, every country will have their own regulations, which should be considered when deciding on model selection.

***Costs***—the cost of large animal model generation and upkeep is considerable but the benefits for appropriately bridging the translation gap leading to effective therapies cannot be underestimated. For example, when modelling a neurological condition that correlates with advanced age, the lifespan of large animal models is more readily suited to late onset phenotypes due to their longevity. However, such model systems require significant financial support for their long-term maintenance, assessment, and sample collection.

The real term cost of large animal model generation is in the region of 25× greater than the cost to create a bespoke rodent model [10,100] and may only generate a handful of founders. Founders will then need to be fully characterised to understand if the engineered disease model has recapitulated phenotypic traits observed in patients. Investment is vital; a successful model can become cost effective by decreasing time from drug trial to regulatory approval, and it increases confidence that therapy or interventions have a better chance of translating to humans [97,101].

***Practicalities***—if using multiple cohorts for longitudinal studies, practical considerations regarding the number of offspring that can be produced from a species, and seasonality of breeding, should be noted. For example, pigs can have two litters per year independent of the season, with each litter comprising 10–12 piglets, but this is generally lower in minipigs 6–8 [102]. By contrast, sheep are typically seasonal breeders and have in the region of 1–3 lambs per year, although some breeds, e.g., Poll Dorsets can lamb up to three times in a 2-year period. Importantly, when selecting a specific species in which to develop a model, it is essential to consider the practicalities of carrying out specific neurological and neuroimaging assessments in that large animal model. For example, certain breeds of sheep are not ideal for Magnetic Resonance (MR)-based neuroimaging and/or surgical delivery of therapeutics due to the presence of horns that prevent application of a suitable head coil or optimal positioning of a craniotomy, respectively. Unless well-trained and habituated in advance, pigs are not amenable to clinical assessment of certain facial sensation parameters, due to fear responses or aggression. Viral or pharmaceutical delivery trials can be successful in pigs if the condition they are modelling presents at young ages, but are limited in adults (excluding mini pigs) due to their gross size and weight presenting challenges to imaging, (e.g., manoeuvring into a scanner) (see Table 2). Finally, behavioural assessment requires experience in recognising and accounting for behaviours that are normal/abnormal for the species within the environment in which they are being tested.

In summary, there is a clear need for larger animal model systems to bridge translational gaps and build on existing work using simpler model systems; however, as with any experimental paradigm, there are a number of considerations which must be taken into account when planning longer term characterisation and interventional assessments.

## 3. Clinical Assessment

In order to fully diagnose and/or characterise any neurological condition, it is imperative to assess the clinical signs the patient is exhibiting, and to provide evidence of location, severity, and progression of disease [11]. Similar assessment criteria are applicable to animals which model any given disorder, but aspects of the exam will need to be tailored to the species of interest, and cognitive testing must be limited to the objective analyses currently available in veterinary medicine. In essence, symptomatic information can only be obtained from and communicated by patients. It is prudent to assume that some impacts on life quality will be missed in veterinary species simply because we are unable to recognise them, and this is an important welfare consideration.

Understanding exactly what examinations can be undertaken in different species and their comparative translation from to human patients is fundamental for obtaining robust characterisation of a human neurological disease model. It is good practice to use a system that gives an appropriate range of scores to easily differentiate severity and give margins for improvement or decline. For instance, depending on the expected neurological progression, a scale of 0–8 is likely more appropriate than scoring 0–3 (0 = normal to 3 = severe). Furthermore, a more comprehensive scoring system allows for subtle changes to be recorded with the caveat that this would be observer-dependent and continuity by consistently using the same scorer is preferred to maintain an equivalent level of subjectivity. Interpretation of reflex tests is best approached more cautiously, limiting the range of scores to what can be observed objectively and reproducibly in the given species. Pilot testing should be performed to determine the expected intra- and inter-observer variability in clinical scoring before embarking on any experimental study with live animals.

Neurological examinations are designed primarily to detect the presence of neurological dysfunction and then to determine the neuroanatomical location(s) that has been affected by damage or disease [103,104]. In the clinical setting, neurolocalisation is established prior to generating a list of differential diagnoses (Table 1), primarily because different diseases can affect different parts of the nervous system in a focal, multifocal, or even diffuse manner [105]. For these examinations to have meaningful results, they should be undertaken by an experienced veterinarian to score the experimental animals, and appropriate age-matched controls. This may require a specialist in veterinary neurology and/or a veterinarian that specialises in the species under test (ideally both). Some of the earliest signs of disease may also be detected by the personnel who manage the animals on a daily basis; their experience and observations are invaluable and should be documented carefully. The animals should be handled often enough that routine examinations are tolerated as a non-stressful experience, but not in such a way that habitual responses to touch mask true deficits in other sensory modalities. It is good practice for the personnel involved in husbandry and the examiner to be ‘blinded’ to the disease status of the animal, where possible. If behavioural testing of each animal is needed away from the flock/herd, the presence of a familiar handler is paramount [106].

Whilst it is best practice to perform the most aversive aspects of the neurological exam last, the order in which tests are presented below is not intended to dictate the order in which they should be carried out—this will vary by species, experimental design, and preference of the examiner. The exact sequence of tests is less important than being systematic, comprehensive, and keeping the sequence consistent across animals, and across repeated measures for an animal, in any given experiment. It is highly recommended to document key elements of the examination (particularly behaviour, posture, and gait) using a standardised video recording protocol. Alongside documenting progression of neurological signs, video enables additional ‘hands off’ assessment by multiple observers who can be blinded to disease status and stage; moreover, slowing of the viewing speed permits subjective (and sometimes objective) evaluation of subtle deficits.

Here, we describe specific assessments to determine the neuroanatomical location of of dysfunction, highlighting some practical aspects most relevant to livestock species. As with any veterinary species, ‘hands off’ evaluations should be performed prior to tests that involve progressively more handling.

### 3.1. Mental Status and Behaviour

This initial assessment can provide valuable additional information to the owner/handler’s management history. The examiner should observe the animal in situ, first from a distance (without being disturbed) and then closer (without being handled) in order to judge how it responds and interacts with everyday environmental stimuli [13,22]. The level and quality of consciousness should be assessed, with the level of consciousness being classified as alert, obtunded, stuporous, or comatose (in order of increasing deficit) [12]. An animal which is bright and/or quiet, alert, and attentive to its environment would require an intact cerebrum and ascending reticular activating system, which extends over the entire length of the brainstem to activate the cortex [12,107]. Unusual interaction or dulled responses to the surroundings therefore indicate a lesion to the forebrain and/or brainstem [20]. Notably, large chronic forebrain lesions in livestock can be clinically silent and may therefore correlate poorly with neuroimaging results [13]. The quality or ‘content’ of consciousness refers to inappropriate responses to stimuli or abnormal behaviours that can occur even in the presence of a high level of consciousness [12].

Head pressing or ‘getting stuck in corners’, compulsive wandering, head/neck turning and circling (typically towards the worst affected side), and signs associated with blindness (such as tripping over or bumping into obstacles), suggest problems associated with the thalamocortex, often with a metabolic, traumatic, toxic, or inflammatory cause [13,20,104]. Seizure activity, whether generalised or focal, establishes forebrain dysfunction and typically comprises a humane end point. Sometimes seizure activity can be elicited by auditory or tactile stimuli [13]. In the absence of classic generalised tonic–clonic activity with autonomic signs, however, other types of seizure activity can be challenging to detect and to differentiate from stereotypy, syncope, movement disorders, or sleep disorders without continuous video monitoring and electroencephalography.

### 3.2. Posture and Stance

Posture is evaluated systematically in three anatomical regions: (1) head, (2) body, and (3) limbs [22]. Careful assessment of an animal’s muscle tone and condition in each region (1, 2, and 3), identifying any muscle asymmetry and/or atrophy through observation and palpation, can provide clues for further investigation [108]. If tolerated, straightening the head and neck along the midline can help with assessment of any head/neck deviation, for which musculoskeletal problems must be ruled out [13]. Otherwise, abnormalities in attitude or positioning of the head (e.g., head tilting) and/or limbs in relation to the body (e.g., leaning, broad-based stance, one or more limbs are not situated in their normal axial position) indicate a lesion of the vestibular system [105]. The vestibular system’s primary function is maintenance of balance and positioning of the eyes, neck, body, and limbs relative to the movement of the head [14].

Importantly, a head tilt must be distinguished from a head turn; the former specifies a lesion affecting the vestibular system, whereas the latter is a sign of forebrain dysfunction. It is possible however for a severe vestibular lesion to produce a head tilt together with a head and neck turn [13]. Severe forebrain disease, myotonia, and visual loss can all result in a ‘star-gazing’ posture with opisthotonos of the head and neck [13]. Star-gazing may be part of decerebrate rigidity, but other acute neurological postures such as decerebellate rigidity, and Schiff–Sherrington posture are far more common in small companion animals [12]. Kyphosis, lordosis, scoliosis, torticollis, and abnormal limb postures such as palmigrade or plantigrade stance are rarely observed in livestock beyond congenital disorders and acquired trauma [13].

### 3.3. Gait Analysis

Gait analysis can help track the effect of disease progression on vestibular, motor, and proprioceptive function as well as visual capacity [109]. Along with mentation, gait should be initially assessed (within reason) in the animal’s normal environment prior to more hands-on elements of the neurological exam. That said, for some species (especially equids), the most informative gait assessment usually results from a combination of examiner-led maneuvers using halter and lead rope, with and without tests of strength, balance, and proprioception through tail pulling and circling [13]. Nonetheless, key first steps in all cases are to establish which (if any) limbs are abnormal and to rule out non-neurological gait deficits such lameness due to orthopaedic disease. As a standard, observation should be made of the animal walking in a straight line, viewed from the side, front, and back, ideally with video recording. Stride length, trajectory, pattern of foot placement, and any evidence of toe dragging or head nodding must be carefully assessed, noting that a short-strided gait can arise from pain, weakness (neurogenic or non-neurogenic in origin), or incoordination. Ataxia, by definition, comprises incoordinated (typically irregular and unpredictable) motor activity that does not result from weakness or musculoskeletal issues, and is therefore neurological in origin [12]. It can affect movements of the head, neck, trunk, limbs, eyes, or a combination thereof, and involves elements of hypermetria, hypometria or both (dysmetria) [13]. Ataxia can indicate a lesion affecting the brain stem, cerebellum, spinal cord or peripheral nerves [21,22] and is categorised as either vestibular, cerebellar, or proprioceptive.

*Vestibular ataxia* typically includes a hypermetric, wide-based, staggering gait with exaggerated extension of the limbs and leaning and falling towards the side of the lesion as a result of relative hypertonia on the unaffected side (in the case of unilateral vestibular disease) [20,105]. Mild signs are exacerbated when blindfolding the animal. Unilateral vestibular lesions usually also produce a head tilt and abnormal nystagmus (see below). Bilateral vestibular dysfunction (central or peripheral) is characterised by swinging head and neck ataxia in the absence of a clear head tilt, and a crouched posture with reluctance to move [12,13]. *Cerebellar ataxia* features alterations in the rate, range, and force of movement (a dysfunction of subconscious proprioception) manifesting in dysmetria (classically a jerky, hypermetric, high-stepping gait along with intention tremor or titubation of the head and trunk) [13]. This type of ataxia is caused by a lesion of the cerebellum (sometimes with associated deficits in the menace response) or spinocerebellar tracts [12]; affected animals may also adopt a broad-based stance but typically show no signs of weakness (paresis). Lastly, *sensory* or *proprioceptive (spinal) ataxia* results from lesions affecting ascending pathways for subconscious proprioception in the spinal cord and brainstem and is best described as an irregularly irregular (and therefore highly unpredictable) gait [13]. Scuffing, toe dragging, or knuckling with hypometria in the thoracic limbs, alongside hyperflexion of the pelvic limbs is often observed, together with a wide-based stance, a delayed-onset stride with swaying or floating swing phase, and abduction or adduction during foot placement [13]. Circumduction, crossing of the limbs, or stepping on the opposite foot may also be seen. Abrupt changes in direction, circling, backing the animal up, or walking it up or down a slope with the head elevated can help identify proprioceptive ataxia, which can be very subtle in the early stages of disease [20]. Proprioceptive ataxia is often accompanied by varying levels of paresis.

*Paresis* or weakness involves partial loss of voluntary movement; there may be reduced ability to initiate a gait, to maintain a posture, to support body weight, and/or to resist gravity [12,13]. Typically, a decreased rate or range of motion will be observed (hypoflexion and hypometria), sometimes manifesting as a short-strided bunny hopping gait, alongside increased fatigability, hypotonia, low head and neck carriage, and sometimes postural tremor [12]. A ‘two-engine’ gait with short strides in the thoracic limbs and long strides in the pelvic limbs is characteristic of, but not pathognomonic for, a C6-T2 spinal cord lesion involving grey and white matter [13]. Paralysis (plegia) is defined as complete loss of voluntary movement.

Sophisticated electronic systems such as GAITRite or Zeno walkway from ProtoKinetics have been used to quantify gait in to infer disease and treatment status in a broad range of conditions from Parkinson’s disease, multiple sclerosis, and stroke, to residual deficits following complex orthopaedic surgeries [109,110,111]. This system, and similar derivatives, have been adopted in some large animal studies which use a 5 m electronic walkway to measure cadence, speed, stride length and width as well as individual pad pressure [96,112] in a similar fashion to the catwalk systems that are commonly used in rodents [113]. GAITrite was able to detect subtle gait changes in CLN5 disease sheep but not the ovine model of Huntington Disease [106].

If gait analysis systems are not readily available, gait can be measured by using more crude techniques developed in rodent models in which the feet are painted and the animals walk along a length of paper [114]. It is advisable to paint the front and hind feet in different colours to easily discriminate horizontal distances between the front to front and hind to hind, and also, the stride length and pattern (Figure 1). This manual form of gait analysis will be limited to species in which the size of the animal permits painting of the underside of the feet. Animals should also be habituated to the process of walking on the paper prior to any testing occurring to ensure a ‘normal’ gait is recorded.

### 3.4. Cranial Nerve (CN) Tests

It should be noted that CN I and XI cannot be adequately tested in livestock species in routine practice [20]. Whilst normal olfactory nerve (CN I) function may be suggested by the sniffing response of a blindfolded animal to the presence of a food item near the nostril, this almost certainly invokes additional stimulation of sensory afferents of the trigeminal nerve (CN V) supplying the nasal mucosa [12,13]. The spinal accessory branch of the accessory nerve (CN XI) provides motor innervation to trapezius and cranial sternocephalicus muscles which are rarely atrophied in isolation, and thus lesions are difficult to detect [13] (Figure 2).

#### 3.4.1. Pupillary Size and Symmetry

The optic nerve (CN II) can be assessed through evaluating pupil size and symmetry, performing pupillary light reflex and menace response tests, and by monitoring for behaviours consistent with visual loss. Pupillary size should be noted bilaterally and any anisocoria (asymmetrical pupil size) documented [12]. Subtle anisocoria is often of no significance and is easily missed in livestock because the lateral globe positioning makes it difficult to examine both pupils simultaneously. Looking through a direct ophthalmoscope, an arms-length away from the animal’s head, on a low light setting, is ideal for assessing resting pupil size. Healthy pupillary size is ultimately determined by a balance between parasympathetic oculomotor nerve (CN III) input to pupillary constrictor muscles via the ciliary ganglion, and sympathetic innervation of the pupillary dilator muscles arriving via the cranial cervical ganglion [13]. Anisocoria has many potential causes, however, some of which are non-neurological in origin, and the reader is referred to a detailed discussion of this elsewhere [14].

#### 3.4.2. Pupillary Light Reflex (PLR)

The PLR (Figure 2A) tests involuntary aspects of the light-responsive pathways, i.e., those that require no input from higher centres in the brain. The afferent pathway comprises the retina, optic nerve (CN II), optic chiasm, optic tracts lateral and dorsal to the thalamus, pretectal nuclei, and finally the parasympathetic motor nuclei of the midbrain, from where the efferent output is mediated by parasympathetic fibres of CN III [13]. The examiner tests the PLR in each eye by shining a bright light in a left to right direction, toward the maximum density of rods and cones [20]. This causes the pupil of the illuminated eye to constrict (the direct PLR). At the same time, this should cause constriction of the contralateral pupil (the indirect or consensual PLR; [115]. It should be noted that ruminant pupil constriction during the PLR is much slower than that of dogs and cats, but is generally faster than that in horses [14,22]. The result of the direct reflex is normally greater than that of the consensual reflex because most optic nerve axons decussate at the optic chiasm and most pretectal nuclear axons cross back again at the caudal commissure [14]. This differential effect is most notable in livestock species and horses because a greater proportion of fibres (80–90%) cross over at the optic chiasm [14]. Observing the PLR in both eyes simultaneously is challenging in livestock and so the swinging light test may be the most practical approach [13]. Diseases which progressively affect CNs II or III will typically lead to a loss of vision before a loss of the PLR [14]. The oculomotor nerves are vulnerable to pressure-induced injury in the context of diffuse brain swelling and space-occupying lesions of the forebrain [13].

#### 3.4.3. Dazzle Reflex

During PLR testing, animals sometimes partially blink or squint, retract the globe (with protrusion of the nictitating membrane), and/or avert the head. This is known as the dazzle reflex and is mediated via the retina, optic nerve, (CN II), rostral colliculus and/or supraoptic nucleus of the hypothalamus, facial nucleus, and facial nerve (CN VII); [14,116,117]. Since all processing of this photo-aversion is subcortical, the result is entirely involuntary and is typically only observed with the intense stimulation of a very bright light source. It may however be stronger in photophobic animals, or those who are acclimatised to the dark [116,118]. The dazzle reflex may be particularly useful when observation of the pupils during PLR testing is precluded by severe corneal oedema or hyphema [117]. More recent research suggests a contributory role for intrinsically photosensitive retinal ganglion cells (ipRGCs) in the dazzle reflex [118]. ipRGCs also regulate light responses over longer timescales, projecting directly via the retinohypothalamic tract to the suprachiasmatic nucleus where they help regulate circadian entrainment [119]. Given the increasing evidence of circadian disruption in chronic brain disease [120], it may be prudent to include dazzle reflex testing routinely in neuro-ophthalmic assessments of animal models of these disorders.

#### 3.4.4. Menace Response

The menace response tests vision through evaluating the ipsilateral optic (CN II) and facial (CN VII) cranial nerves, as well as the ipsilateral cerebellum and the contralateral forebrain (thalamocortical system) [12]. It requires integrity of the entire visual pathway including sensory and central processing (retina, optic nerve, optic chiasm, optic tract, lateral geniculate nucleus of the thalamus, optic radiation, and occipital cortex), as wells as integration and output pathways via the cerebellum and facial nerve [14,20]. A normal menace response therefore confirms that all the above structures are functional, but a weak or absent menace response poorly isolates lesion location. It is recommended to first test the palpebral and pupillary light reflexes to ensure the animal has capacity to blink and respond to light, respectively, before testing the menace response [12]. Each side should be tested in isolation, covering the eye that is not being examined—although this is less important in livestock species in which the eyes are positioned laterally [20]. The animal should be positioned approximately 30–50 cm away from the examiner and a threatening gesture made by the examiner’s finger being rapidly directed towards the animal’s eye, testing both nasal and temporal regions of the retina [13,20,22]. Generation of air currents onto, and any direct contact with, the ocular surface must be avoided to prevent invocation of the corneal reflex via stimulation of the ophthalmic branch of the trigeminal nerve (CN V). Similarly, touching any facial hairs or eyelashes will erroneously elicit the palpebral reflex (Figure 2B) [14]. The normal response is to blink, but this is a learned response which is not present in dogs and cats younger than 12 weeks of age and in foals, piglets, calves, and lambs younger than 7 to 10 days old [14,115]. Animals with a facial nerve deficit may instead retract the globe (noted by protrusion of the third eyelid) and/or withdraw the head if the remainder of the pathway is intact [12].

It is crucial, however, to account for interspecies differences to allow appropriate comparisons to be made. For example, prey species such as sheep with laterally directed axis of vision and wider field of view have comparatively smaller binocular visual coverage and larger monocular visual coverage than humans, resulting from varying degrees of decussation at the optic chiasm, and thus different amounts of binocular cortical representation in the primary visual cortex [121]. Generally, as binocular capacity increases, decussation at the chiasm decreases; in primates, the extent of decussation is just above 50%, whilst in livestock it is 80–90% [14]. Sensitivity to motion in peripheral vision combined with hyperopia in the central field must be considered when comparing response to visual stimuli with humans.

#### 3.4.5. Eye Position

CNs III, IV, and VI together with the vestibular system, determine the position of the eye in the orbit by providing motor efferents to the extraocular muscles. With the head at rest, both eyes should point in the same direction. If this is not the case, a strabismus must be recorded. Abducens nerve (CN VI) lesions will cause a medial strabismus (abnormal angling of the eye towards the nose) and will potentially affect globe retraction; a deficit which may be noted during corneal reflex testing. Trochlear nerve (CN IV) deficits will produce a rotational strabismus (with dorsal deviation of the medial aspect of the pupil in ruminants and horses), and oculomotor nerve (CN III) deficits may produce a ventrolateral strabismus [14]. A positional strabismus may be unmasked by moving the head into different positions [12], a manoeuvre which may also elicit a positional nystagmus. Whilst dorsomedial rotation of the eyeball is characteristic of diffuse brain disorders in ruminants and remains controversial in its neurolocalisation (some consider CN IV dysfunction); positional strabismus and/or positional nystagmus are almost always due to vestibular dysfunction [13].

#### 3.4.6. Vestibulo–Ocular Reflex (VOR)

Together with the vestibular portion of CN VIII; CNs III, IV, and VI are further responsible for conjugate eye movements [20]. Nystagmus is involuntary oscillation of the eyeball [14]. Normal eye movement can be tested by eliciting physiological nystagmus, also known as the VOR or oculocephalic reflex. Both eyes are observed whilst moving the head repeatedly in a horizontal and then vertical direction [12]. Normal physiologic nystagmus has a fast phase in the direction of head movement and the extent and direction of gaze in both eyes should be the same—towards the direction of head movement [115]. Note that in unaffected ruminants and horses the eyeballs tend to maintain a horizontal gaze and thus move ventrally in the orbits when the head and neck are extended [13,14]. The eyes should stop moving as soon as the head is at rest; continued movement or the presence of nystagmus at rest is abnormal (pathological or spontaneous nystagmus) and indicates a vestibular disorder [20]. A positional nystagmus is always abnormal and refers to one that is only observed when the head is elevated, or changes direction when the head is elevated. In the case of a peripheral vestibular problem, the fast phase of the nystagmus is directed away from the side of the lesion and the plane of the oscillation is horizontal or rotatory [14]. A consistently vertical nystagmus, or a nystagmus that changes with different head positions usually indicates a central vestibular problem [14].

#### 3.4.7. Corneal and Palpebral Reflexes

The corneal reflex involves stimulation of the ophthalmic branch of the trigeminal nerve (CN V) with the efferent mediated by the abducens nerve (CN VI). The test should be performed carefully (to avoid damage to the ocular surface), either by palpation through closed eyelids, or by gently touching the cornea in a region away from the pupil aperture using a cotton bud moistened with saline [12,13]. The normal result is retraction of the globe, the fullness of which may depend on the functioning of all extraocular muscles in livestock and horses [13]. The palpebral reflex (Figure 2B) assesses the sensory function of the ophthalmic and maxillary branches of CN V, via stimulation of the medial and lateral canthi of the eye, respectively. The motor efferent is via CN VII leading to wrinkling of the face and blinking [14]. A full assessment of facial sensation (but not perception of it) would include blunt stimulation of the head and face in all key areas of sensory innervation provided by the major branches of CN V [13]. For example, touching the base of the ear to stimulate the sensory portion of the mandibular branch of CN V should elicit reflexive eyelid closure ipsilaterally by CN VII [20].

#### 3.4.8. Nasal Sensation

Evaluation of nasal sensitivity assesses if there are abnormalities in the ipsilateral ophthalmic or maxillary branch of the trigeminal nerve (CN V), or in the nociceptive pathway to the contralateral cerebral hemisphere [14]. A normal response is withdrawal of the head. The examiner would assess sensitivity by touching the nasal septum for the ophthalmic branch (Figure 2C) or the external parts of the nostril for the maxillary branch of CN V [12].

#### 3.4.9. Facial Symmetry

The examiner should look at the ears, eyelids, lip commissures, and nostrils to check for symmetry (Figure 2D) [20,21,115]. A narrow palpebral fissure (loss of function of the levator anguli oculi medialis muscle) or any other asymmetry such as unilateral drooping of the ear or lip, indicates facial nerve (CN VII) motor dysfunction [14]. In horses and sheep, deviation of the nose and upper lip toward the normal side will be evident, whereas in cattle, this deviation is prevented by the large planum nasolabiale [14]. Loss of buccinator tone may lead to residual food in the cheek pouches [13,122]. CN VII is a mixed nerve with parasympathetic fibres that supply mandibular and sublingual salivary glands and others that mediate lacrimation—this can be evaluated using the Schirmer tear test [12]. Additional motor fibres of CN VII supply the caudal belly of the digastricus muscle and the stapedius muscle of the middle ear, whilst afferent CN VII fibres provide taste sensation to the rostral two-thirds of the tongue. None of these functions are readily tested in the neurological exam.

#### 3.4.10. Jaw Tone

The examiner should test the resistance of the animal to having their mouth opened. A flaccid or dropped jaw with inability to chew indicates total bilateral loss of trigeminal nerve (CN V) motor function (mandibular branch) [13,115]. Tongue protrusion and drooling may be observed, followed by atrophy of the muscles of mastication [13].

#### 3.4.11. Startle Response

This test assesses responses of CN VIII (most likely the vestibular portion according to recent studies) to a loud and unexpected noise, with resultant body and limb movements mediated by the tectospinal tract [13]. A true startle is thus considered as a reflexive, subconscious response. As prey species, however, livestock are generally very sensitive to sound, and the lack of any response to a standard auditory cue might indicate a deficit in the cochlea of the inner ear, or other region of the auditory pathway via CN VIII and into the brainstem, from where sound information projects bilaterally up to the midbrain, thalamus, and auditory cortex [13,115]. As with humans and other domestic animals, these pathways are best tested using the brainstem auditory evoked response (BAER).

#### 3.4.12. Swallow (Gag) Reflex

Abnormalities in swallowing are known as dysphagia and could be due to lesions in the glossopharyngeal (CN IX), vagus (CN X) and/or accessory (CN XI) nerves which together provide sensory and motor innervation to the pharynx and larynx [13]. In order to determine the efficacy of the animal’s swallow response, the examiner would insert a finger caudally into the mouth as far as possible until the reflex is initiated. With ruminant pharyngeal paralysis, food, water, and saliva may be noted in the nostrils and mouth [13]. Rubbing the ventral larynx can also initiate the swallow reflex. Any change in voice/vocalisation should also be noted and could be caused by damage to CN IX and/or X [20,104,123].

#### 3.4.13. Tongue Tone

Motor control of the tongue is governed by the hypoglossal nerve, CN XII. To assess CN XII function, the examiner should open the animal’s mouth to observe the position and movement of the tongue and assess for any atrophy [20]. The tongue can also be grasped to assess for a normal level of strength/resistance [115]. A unilateral lesion will lead to lateral deviation of the tongue, the direction of which may depend on the stage and extent of dysfunction. Bilateral dysfunction will impair prehension and swallowing [13].

### 3.5. Spinal Reflex Tests

#### 3.5.1. Patellar Reflex (a Monosynaptic, Myotatic Reflex)

This tests the reflex arc comprising sensory and motor fibres of the femoral nerve, and primarily spinal cord segments L4 and L5 [21]. The test can be performed in recumbent animals or those that are amenable to being positioned in lateral recumbency or turned onto their back and held in a seated position (e.g., small ruminants; [115]). Each pelvic limb should be tested when uppermost and when dependent and the tested limb should be supported in a partially flexed position with slight tension in the muscle [13]. The middle patellar ligament is then tapped with a patellar hammer and the normal response is swift contraction of the quadriceps muscle with stifle extension. This action must be carried out with precision to avoid striking the muscle tissue or bone giving rise to erroneous results (Figure 2E).

#### 3.5.2. Extensor Carpi Radialis (ECR) Reflex

The ECR reflex tests the integrity of the radial nerve and spinal cord segments C7-T1. The antebrachium is held, keeping the elbow flexed, and the belly of the carpi radialis muscle is sharply struck just distal to the elbow [12]. Normally, contraction of the ECR results in an extension of the carpus [21]. This may be a muscular response rather than a true myotatic reflex as it has been observed in dogs with a transected radial nerve; the results of this test may not be reliable in adult livestock [12,13].

#### 3.5.3. Withdrawal (Flexor) Reflexes

The withdrawal reflexes test the function of: (1) C6-T2 spinal cord segments and the musculocutaneous, axillary, median, radial, and ulnar nerves in the thoracic limbs and (2) L5-S3 spinal cord segments and the sciatic nerve in the pelvic limbs [14,20]. The animal is assessed in lateral recumbency and, using a pair of artery forceps, the examiner pinches the interdigital skin between the toe(s) in the paw or hoof (Figure 2F). The normal response is withdrawal of the thoracic limb via flexion of the shoulder, elbow, and carpus, or withdrawal of the pelvic limb via flexion of the hip, stifle, and hock. A neurologically normal animal will also display a behavioural response to this noxious stimulus, since it tests the nociceptive pathway from spinal cord to cerebrum [13].

#### 3.5.4. Perineal (Anal) Reflex

The perineal reflex is performed by lifting the tail and gently prodding the skin on each side of the perineum with a pen or cotton swab (Figure 2G). A normal reaction includes reflex contraction of the anal sphincter and clamping down of the tail. The sensory arm of the reflex involves the perineal branches of the pudendal nerve from S1–S3. Contraction of the anal sphincter is mediated by caudal rectal branches of the pudendal nerve, and tail flexion by the sacral and caudal segments and nerves, S1–Ca [13]. In adult cows, gentle stroking of the anus typically causes reflex tail extension a micturition reflex. Sensory perception of these stimuli may still be present even when the segmental reflex functions are lost [13].

#### 3.5.5. Cutaneous Trunci Reflex

The cutaneous trunci reflex assesses the sensory integrity of all dermatomes over the thoracolumbar vertebral column, spinal cord segments C8-T1, and the lateral thoracic nerve running to the cutaneous trunci muscle of the flank [12]. The examiner would elicit a stimulus by pinching the skin on both sides of the thoracolumbar region which should make the cutaneous trunci contract bilaterally (Figure 2H) [115]. The reflex may not be elicited caudally to the level of a spinal cord lesion but should be intact cranially to it and the examiner may need to carry out multiple stimulations for the animal to respond normally [20,21].

### 3.6. Postural Reactions

Postural reactions primarily help to identify subtle proprioceptive and motor system lesions in the absence of obvious gait deficits [13,124]. These tests will also evaluate the visual system and the cerebellum [115] but are more challenging to perform in large animals such as horses and pigs [14]. Although not all of these tests inform on the anatomical location of a lesion, they do help build a profile of neurological dysfunction with abnormalities detected in these assessments not necessarily observed in gait analysis [124].

#### 3.6.1. Knuckling/Paw or Hoof Placement Test (Proprioceptive Positioning)

A neurologically normal animal should immediately correct the abnormal positioning of a foot if it has the strength and attention to do so [14]. In a properly supported animal (which may be difficult in most large species), each paw or hoof can be turned over by the examiner so that the incorrect (dorsal) surface touches the ground (Figure 2I).

#### 3.6.2. Hop Test

Hopping is a sensitive test for subtle weakness or asymmetry [12]; it simultaneously assesses the function of the visual cortex, cerebellum, and spinal cord and is often considered the most reliable postural reaction test [14]. The examiner should pick up and support one limb, so that three out of the four limbs are on the ground, and move the animal laterally sidewards (Figure 2J). This should be carried out for each limb. The normal animal will hop quickly and smoothly in the direction in which it is being moved in order to resist gravity and maintain balance [14]. Cerebral lesions produce contralateral deficits, whilst brainstem and spinal cord lesions produce ipsilateral deficits [13].

#### 3.6.3. Step/Visual and Tactile Placing Test

As for Section 3.6.1, the animal should be supported by the examiner and/or assistant throughout the assessment. With (tactile) and without (visual) blindfolding of the animal, the foot can be crossed over its contralateral counterpart or slowly abducted by the examiner until the animal lifts the foot and correctly places it underneath its body (Figure 2K). Use of a sliding feed sack beneath the foot can help avoid the invocation of a withdrawal reflex through manual handling of the limb. It should be noted that the utility and interpretation of Section 3.6.1 and Section 3.6.3 in livestock and horses remains controversial; spontaneous abnormal positioning of the limb(s) after abruptly stopping the movement of a large animal may be a more informative test of conscious proprioception [13].

#### 3.6.4. Push Test

The push test assesses strength, balance, coordination, and postural stability and is a test commonly carried out in patients suspected of having Parkinson disease [125]. The examiner should support the animal and push against its body and should feel there is resistance in the animal to being pushed (Figure 2L). In horses, tail pulling in stance and during ambulation is more often employed to detect extensor weakness and proprioceptive deficits [13].

### 3.7. Pain Perception (Nociception)

Assessing an animal’s response to pain (nociception) is important to help pinpoint areas that may have undergone mechanical, soft tissue, or neurological injury evoking an emotional response. This reaction may be in response to a number of different stimuli and is a protective behaviour to alert an animal to danger or protect it from further injury by, for instance, not bearing weight on a damaged limb [126,127]. Other clinical signs of pain in an animal may include loss of appetite, withdrawn demeanour, unwillingness to work, and vocalisation [126]. Additional neurological pain can present as hypersensitivity to touch or temperature. A behavioural response elicited during withdrawal reflex testing is a good indicator that sensory afferents in the peripheral nerves, spinal cord, and pathways ascending to the brainstem and forebrain are intact [128].

### 3.8. Cognitive Assessment Trials

Assessing cognitive decline in patients suffering from neurodegenerative disease is well established with tests such as the dementia detection test (DemTect) or Montreal Cognitive Assessment (MoCA) that rely on language for communication [129,130]. This poses an additional challenge for assessment in animals, but tests have been devised that inform on deficits of memory and spatial awareness in animals. Here, we list two tests that have been successfully implemented in large animal models, building upon the clinical data acquired.

#### 3.8.1. Maze Testing

Maze tests help to assess an animal’s memory and spatial awareness, but can also be used to unmask more subtle visual loss, using non-aromatic, non-auditory visual cues [13]. Due to the preference of most livestock species to remain in herds or flocks, separation from their pen mates can be stressful but re-integration into a familiar group of conspecifics can also serve as a useful reward. Visual awareness of the pen mate/owner’s location at the end of a maze or course is often enough motivation for the animal to navigate through the maze alone. A good example of a successful use of a maze trial was published by Mitchell et al. [131], who have shown that sheep with naturally occurring CLN5 disease progressively lose the ability to quickly navigate the same maze over time due to disease advancement. Moreover, this assessment was used to aid determination of the efficacy of gene therapy treatment and clearly identified treated individuals as ‘out-performing’ those that had not been treated.

#### 3.8.2. Executive Function

Executive function skills which allow us to focus, memorise, and multitask are vital cognitive functions which also allow us to plan, organise, and reason [132]. These skills are diminished with progressive neurodegeneration associated with normal healthy ageing but are markedly lost in a variety of other disorders, including, but not limited to, Alzheimer’s, Huntington’s, and motor neuron disease [133,134,135]. Just as these tests are crucial to assess advancement of disease in patients it is important to devise similar exams for large animal models of these diseases. Morton and Avanzo successfully implemented a range of tests in sheep in order to assess executive function which included discrimination learning, retention, and reversal learning with the latter used to determine the functional integrity of striatum and pre-frontal cortex in Huntington patients [134]. Follow-up studies have also objectively measured stop-signal reaction time and facial recognition in this species [136].

## 4. In Vivo Imaging Assessment(s)

### 4.1. Central Nervous System (CNS)

Another key tool for diagnosis and assessment of neurological disease is cross-sectional imaging, most commonly computed tomography (CT) and magnetic resonance imaging (MRI) in clinical settings, with additional low volume specific use of positron emission tomography (PET). The clinical neurological assessment can determine lateralisation/localisation within the neuraxis, with additional demographic and temporal features helping to narrow down potential clinical diagnoses [137]. However, imaging is routinely acquired for confirming or refining the differential diagnosis, and for assessing disease progression or response to treatment [138]. As non-invasive techniques, with minimal, quantifiable risks from ionising and non-ionising radiation and high magnetic fields, neuroimaging represents a unique opportunity to study nervous system structure and function in a longitudinal setting allowing for direct clinical translation when using the same imaging platforms with both large animals and humans. Given the extensive knowledge of structure-function relationships in the nervous system, it is possible to relate the imaging phenotype to the clinical assessment, e.g., measuring optic nerve or chiasmatic thickness in anterior visual pathway disturbances, or confirming whether a pelvic limb motor disturbance is arising from the cord or primary motor neocortex. Whilst very important in primates, it should be noted that corticospinal motor pathways do not play a dominant role in initiating voluntary limb and body movement in domestic mammals; thus, large lesions of cerebrocortical motor centres in isolation do not typically produce permanent gait abnormalities in livestock [13]. Figure 3 demonstrates how the imaging component complements and further scrutinises the outcomes from clinical assessment.

### 4.2. Eyes

In addition to the neuro-ophthalmological tests described above, a complete ophthalmic examination may be beneficial, especially if any deficits in the ocular reflexes and responses are detected. The Schirmer Tear Test I can give an indication of basal and reflex tear production; reference values are available for domesticated species [139] Gelatt K. N et al., 1975. Hand-held slit lamp biomicroscopy (e.g., Kowa SL-17) allows detailed examination of the ocular adnexa, cornea, anterior chamber, iris, lens and anterior vitreous, and in most species can be performed with gentle manual restraint only. An indirect ophthalmoscope headset, combined with a 20–40D lens can give an excellent overview of the fundus. The direct ophthalmoscope can be used to closely examine any detected lesions and the optic nerve head at increased magnification but with a smaller field of view. Intraocular pressure measurements are routinely taken in a non-invasive fashion with a handheld rebound tonometer (iCare TonoVet). Additional tests can be performed as required, for example gonioscopy to assess the iridocorneal angle, electroretinography to interrogate the electrical activity of the retina, fluorescein angiography to examine the vascular components and perfusion of the fundus, and ocular ultrasound (including high-resolution ultrasound, contrast-enhanced ultrasound, and ultrasound biomicroscopy) [139] Gelatt N et al. 1975. Some of these examinations require pharmacological pupil dilation, typically with topical tropicamide, but this is animal model dependent. It is important to time examinations carefully as bright light stimulation (e.g., retinal photography or indirect ophthalmoscopy) is likely to impact electroretinography (ERG) results therefore, it is crucial to allow sufficient recovery time with dark adaptation prior to ERG recording [140]. More advanced imaging modalities, such as SD-OCT (spectral domain ocular coherence tomography) and SD-OCT angiography is likely to form a cornerstone of future research, providing real time in vivo analysis of retinal pathology [141]. In vivo confocal microscopy can image corneal innervation (parameters including corneal nerve fibre length, density, branching density, tortuosity coefficient and beadings frequency) and repeated examinations allow demonstration of disease progression if present [142,143].

## 5. Cerebrospinal Fluid (CSF) Analysis

Techniques for collection and standard analysis of CSF from large animals are described in detail elsewhere [13,21,144,145]. Sampling in large domestic species is generally carried out at the lumbosacral space. However, sampling from the atlanto–occipital space is possible but is more challenging, carries greater risk, and requires stable head positioning under general anaesthesia or deep sedation [145]. It may be that the most informative diagnostic samples are obtained as close as possible to the site of pathology, which for most neurological disease models will be the brain and/or spinal cord. In the research context, the need for CSF analysis needs to be weighed up against the risks of sampling for the animal, and it is important to consider whether serial longitudinal samples are really needed for validation of a particular pathological process. Where the disease process is expected to result in raised intracranial pressure (or where there is evidence of such with advanced neuroimaging), CSF should not be sampled in a conscious animal, nor under anaesthesia, unless this is pre-designated as a non-recovery procedure. Whilst 1–2 mL would be sufficient for routine cytological analysis and determination of protein levels [145], up to 1 mL per 5 kg can be safely removed at one time from companion animal patients [12]. In theory, much larger volumes can be safely obtained from livestock species, but even with cryopreservation of excess sample for future analyses, volumes of more than 5–10 mL are unlikely to be warranted. Where analysis is planned only at end stage disease, or at experimental/humane end points, CSF is best obtained immediately postmortem, with no restrictions on sample site or volume. Certain analytes and standard readouts require immediate processing to be of worth [145]. We do however recommend biobanking of excess samples at −80 °C to facilitate future multicentre and multiomics studies, whilst reducing the number of animals required to answer research questions.

## 6. Advanced Functional Readouts and Emerging Interdisciplinary Opportunities

Whilst electrodiagnostic testing is fairly routine in companion animal neurology [12], it is rarely conducted in livestock species except in the research or academic setting. Nevertheless, it is possible to perform, e.g., electromyography (EMG), electroretinography (ERG), nerve conduction velocity (NCV), brainstem auditory evoked response (BAER), and somatosensory evoked potential tests in these species if relevant to the disease being modelled [13,144,145]. High temporal resolution longitudinal assessments of general physiology and neurophysiology are now possible in large animals, enabled by radiotelemetry, non-invasive activity monitors [106], and conscious ambulatory electroencephalography (EEG) [33]. The latter has been able to identify non-convulsive epileptiform activity and early changes in sleep in a sheep model of CLN5 disease [33,146]. Likewise, early changes in metabolism and circadian function have been captured in a sheep model of Huntington’s disease through metabolic profiling of repeated blood samples and actimetry, respectively [147,148,149]. The increasing importance of detecting more subtle deficits in neurodegenerative disorders such as sleep and circadian disruption underpins the attraction to modelling these disorders in diurnal species such as sheep [120,150,151]. Moreover, species–specific behaviours such as flock activity, combined with modern GPS tracking systems, can be exploited to study more complex social interactions in natural environments [106]. The emerging portability of advanced imaging modalities such as optical coherence tomography (OCT) may offer ‘windows’ into the brain without the need for general anaesthesia [152,153]; such opportunities would undoubtedly benefit from veterinary ophthalmology expertise. As multiomics data tools become more accessible for comparative studies, and species–specific resources such as brain atlases [154,155,156,157,158,159,160] continue to develop, there will be few barriers to truly comprehensive clinical phenotyping of large animal models of neurological disease.

## 7. Conclusions

In this paper, we have summarised the most important considerations when developing and validating a large animal model of neurological disease, from species selection, to practical neurological phenotyping and monitoring. Certain aspects of the standard neurological examination can be challenging to perform in certain species. Therefore, it is essential to choose a species in which the key neuroanatomical functions of interest can be objectively tested prior to engineering a disease state that would modify them. Welfare must always be at the forefront of any decision to create an induced model of disease. This is ever more critical for disorders of the nervous system which, by their nature, can have a devastating impact on life quality. A collaborative approach to clinical phenotyping that incorporates the expertise of specialists in veterinary neurology, ophthalmology, anaesthesia, neuroimaging, and production animal medicine can provide valuable insights into the very earliest stages of disease; these are features which are often elusive in patients. Applied sequentially and systematically, clinical assessments and other diagnostic tests can provide vital biomarkers of disease advancement as well as indices of therapeutic efficacy.

## Figures and Tables

**Figure 1 cells-11-02641-f001:**
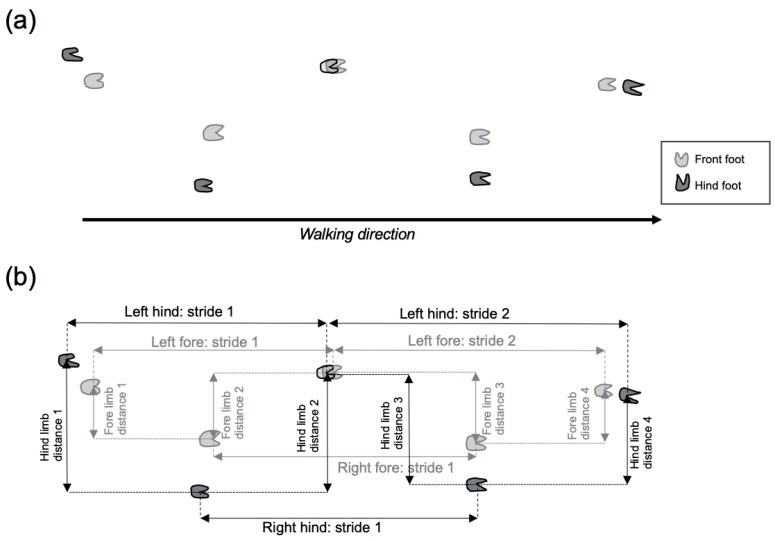
Manual acquisition, analysis, and assessment of gait. Feet are painted in two different colours to differentiate between front and hind limbs (light and dark grey respectively) and animals are made to walk over approximate 5 m length of paper once habituated to the process. (**a**) Overlay of hoof prints from a photograph and (**b**) measurements of stride length (left and right), forelimb and hindlimb distances.

**Figure 2 cells-11-02641-f002:**
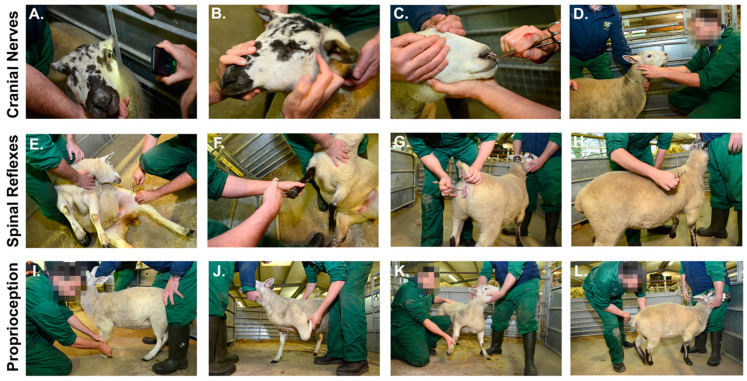
Selection of commonly examined reflexes and reactions to assess neurological function. (**A**–**D**) illustrates cranial nerve assessments with (**A**) pupillary light reflex (PLR), (**B**) palpebral reflex, (**C**) nasal sensation, (**D**) facial symmetry. (**E**–**H**) illustrates spinal reflex exams with (**E**) patellar reflex, (**F**) withdrawal reflex, (**G**) perineal reflex and (**H**) cutaneous trunci reflex. (**I**–**L**) demonstrates proprioceptive positioning with (**I**) knuckling/hoof placement test, (**J**) hop test, (**K**) step/visual and tactile placing test and (**L**) push test.

**Figure 3 cells-11-02641-f003:**
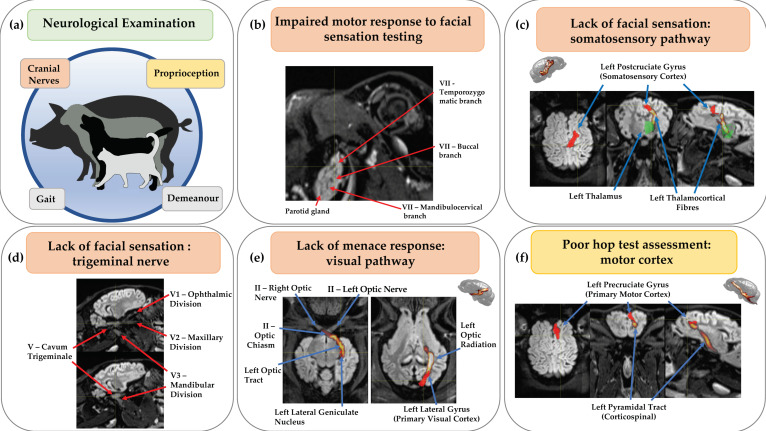
Use of MRI to enable detection of lesions in any part of the reflex pathway, and can be scored or quantified for statistical analysis alongside clinical scoring results. Examples of clinically relevant neuroanatomy from a healthy sheep (Ovis aries). (**a**) Initial neurological assessment by species specific expert identifies abnormalities in cranial nerve and proprioception exams. (**b**) MRI analysis can then determine whether any lesions or abnormalities are present e.g., cranial nerve VII (facial nerve) when facial motor function is impaired in isolation. If facial motor function is intact, but facial sensation is impaired, investigation of the thalamus and somatosensory cortex (**c**) and trigeminal nerve branches (**d**) should be undertaken. (**e**) Abnormalities arising in any of the visual exams can be further scrutinised in the visual pathway to determine if lesions are localised in the optic nerve/tract or chiasm as well as the lateral geniculate nucleus and visual cortex. (**f**) Proprioceptive deficit(s) can also be further investigated with imaging of the spinal cord if the exam pinpoints to a specific region of interest and/or to the motor pathway. Cortical and subcortical regions of interest (ROI; solid red or green) are labelled in the native individual animal using an atlas and template-based approach. Probabilistic tractography is carried out using FSL BEDPOSTX and PROBTRACKX on diffusion-weighted imaging data using cortical and subcortical ROIs. Tract probabilities are shown with a heatmap. The left pyramidal/corticospinal tract, left somatosensory thalamocortical fibres, and left postchiasmatic optic pathway are shown, overlaid on selected orthogonal slices from a fat-saturated 3D FLAIR structural image.

**Table 1 cells-11-02641-t001:** Classification of neurological conditions using VITAMIN D acronym.

Neurological Disease Category	Large Animal Example	Translational Relevance	Reference(s)
	Cerebrovascular accident	Ischaemic stroke	[34]
**V**ascular	Amyloid angiopathy with cerebral haemorrhage in dogs	Haemorrhagic stroke	[35]
	Listeriosis in sheep (infection of the trigeminal nerve and associated brainstem nuclei by *Listeria monocytogenes*	Listeriosis	[36]
	Meningoencephalitis of unknown origin (presumed immune-mediated) in dogs	Autoimmune encephalitis	[37]
		Primary progressive multiple sclerosis	[38]
	Anti-NMDAR encephalitis in dogs and bears	Anti-NMDAR encephalitis aka “brain on fire”	[39,40,41]
**I**nfectious/inflammatory	Toxoplasma-induced encephalitis in utero leading to abortion due to livestock feed contamination with cat faeces	Toxoplasma encephalitis	[42,43]
	Vertebral osteomyelitis in sheep and cattle	Vertebral osteomyelitis	[44]
	Rabies	Rabies	[45]
	West Nile encephalomyelitis in horses	West Nile meningoencephalitis	[46]
	Cysticercosis/taeniasis	Cysticercosis/taeniasis	[47]
	Hydatid disease	Hydatid disease	[48]
	Louping ill	Louping ill	[49]
	Lyme disease	Lyme disease	[50]
	Tick-borne encephalitis	Tick-borne encephalitis	[51]
	Vertebral fracture	Vertebral fracture	[52]
	Peripheral nerve injury	Peripheral nerve injury	[53]
**T**raumatic	Traumatic brain and spinal cord injury	Traumatic brain and spinal cord injury	[54]
	Cerebellar hypoplasia (inherited or infectious origin)	Dandy-Walker syndrome	[55,56]
	Chiari-like malformation	Chiari-malformation	[57]
	Spina bifida	Spina bifida	[58]
Anomalous	Hydrocephalus (infectious, inherited, nutritional or toxic origin)	Hydrocephalus	[59]
	Hypoxic-ischaemic encephalopathy in foals or birth asphyxia in calves	Neonatal hypoxic ischaemic encephalopathy	[60]
	Equine nigropallidal encephalomalacia (chewing disease) causes a Parkonsonian-like phenotype due to ingestion of toxic Russian knapweed	Movement disorder	[54]
**M**etabolic/toxic	Closantel toxicity	Demyelinating disorders	[21,61]
	Tetanus and botulism	Tetanus and botulism	[62]
	Organophosphate intoxication	Organophosphate intoxication	[63]
	Spastic paresis in cattle (hereditary)	Spastic paresis	[64]
**I**diopathic	Epilepsy	Epilepsy	[65,66]
	Narcolepsy/cataplexy	Narcolepsy/cataplexy	[67]
	Brain tumours (intra and extra axial) e.g., glioma, meningioma	Brain tumours (intra and extra axial)	[54]
	Lymphosarcoma leading to spinal ataxia (caused by bovine leukaemia virus)	Spinal lympoma	[68]
**N**eoplastic/nutrional	Swayback (enzootic ataxia) in lambs due to copper deficiency in the ewe during pregnancy	Menkes syndrome	[69]
	Thiamine deficiency in ruminants leading to polioencephalomalacia (cerebrocortical necrosis)	Wernicke-Korsakoff syndrome	[70]
	Prion disease (scrapie in sheep; bovine spongiform encephalopathy in catlle)	Creutzfeld-Jakob disease (CJD), variant CJD (vCJD), Kuru, Gerstmann- Sträussler–Scheinker (GSS)	[71,72]
**D**egenerative	Cerebellar abiotrophy (often hereditary in ruminants)	Cerebellar abiotrophy	[73]
	Lysosomal storage diseases (CLN5 & 6 in sheep, CLN2 in dogs, Sandhoff disease in cats, Tay-Sachs disease)	Lysosomal storage diseases	[74,75,76,77,78]
	Canine degenerative myelopathy (multi-system central and peripheral axonopathy in senior dogs 8 years+)	Amyotrophic lateral sclerosis (ALS) and motor neurone disease (MND)	[79]
	Spinal muscular atrophy in brown swiss calves	Spinal muscular atrophy	[80]

**Table 2 cells-11-02641-t002:** Validity of clinical assessment procedures to determine site-specific neurological deficits in different large animal models.

			Species Tests Are Applicable			
	Test	Target Area	Feline	Canine	Ovine	Porcine *
**Head & Neck / Cranial Nerves**	Menace response	Cranial nerve II & VII, visual & motor cortex, cerebellum	+	+	+	+
	Pupillary light reflex (PLR)	Cranial nerve II & III	+	+	+	+
	Vestibulo-ocular reflex (VOR)	Cranial nerve III, IV, VI & VIII	+	+	+	+
	Startle response	Cranial nerve VIII, auditory cortex	+	+	+	+
	Dazzle reflex	Cranial nerve II & VII	+	+	+	+
	Jaw tone	Cranial nerve V	+	+	+	+
	Nasal sensation	Cranial nerve V, forebrain	+	+	+	+
	Facial sensation	Cranial nerve VII	+	+	+	+
	Swallow reflex	Cranial nerve IX, X, XI	+	+	+	+
	Tongue tone	Cranial nerve XII	+	+	+	+
**Spinal cord & proprioception**	Cutaneous trunci	Spinal cord (afferent T3-L1) (efferent C8/T1)	+	+	+	+
	Step/visual and tactile placing test	Visual system and cerebellum	+	+	+	–
	Knuckling/paw or hoof placement test	Spinal cord forelimb (C1-5) Hindlimb (C6-T2)	+	+	+	–
	Patellar reflex	Spinal cord segments L4 & L5	+	+	+	–
	Hop test	Visual cortex, cerebellum, and spinal cord	+	+	+	–
	Push test	Somatosensory cortex, cerebellum	+	+	+	–
	Withdrawal (flexor) reflex: Forelimb	Spinal cord (C6-T2), lower motor neurons	+	+	+	–
	Withdrawal (flexor) reflex: Hindlimb	Spinal cord (L4-S2), lower motor neurons	+	+	+	–
	Perineal (anal) reflex	Spinal cord S1-3	+	+	+	–

* Minipigs should be treated as ovine only if they have been well-handled.

## Data Availability

Not applicable.

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
