# Peer review of "Modelling Neurological Diseases in Large Animals: Criteria for Model Selection and Clinical Assessment"

_cells, 2022, doi:10.3390/cells11172641_

Round 1

Reviewer 1 Report

The authors summarized the neurological assessment protocols on neuroanatomical dysfunction large animal species, which provides direction and guidance for neurological disease modelling. The review is comprehensive, and the figures are well drawn delivering clear information. Only some minor points need to be clarified.

Table 1: lack of hemorhagic stroke in category "vascular". And lack of AD and PD, which are the main types of neurodegenerative diseases.

Reviewer 2 Report

Cells

Review

Samantha L Eaton et al.

Modelling neurological diseases in large animals: criteria for model selection and clinical assessment

Advances in neuroscience are crucial to keep our aging society healthy. The risk of memory loss and cognitive decline grows as people get older. One in six people will experience depression at some time in their life (according to WHO, APA); with women at all ages affected twice as much to develop neuro-psychiatric disorders. The brain is a very complex organ, and plenty rodent, genetic animal research has given invaluable information that has been transformed into medical knowledge. Until reliable alternative approaches for translational research are available and proven, the use of animal models for basic neuroscience is necessary. In the current review, Eaton et al. favorize the use of larger animals, i.e. livestock, to ‘bridge the translational gap’ to humans. In their overview, the authors summarize and describe the optimal use of e.g. sheep, and categorize neurological conditions following ‘vitamin d’ classifications. The review is very comprehensive including assessment strategies behavior, histology, imaging, and electrodiagnostic that have been used to study livestock for neurological analysis. Authors conclude with an advanced functional readout describing (sheep) disease models such as for Huntington or Epilepsy. It’s a novel, and very extensive review. What’s maybe missing, is a discussion, summarizing paragraph near the reviews end, that compares the authors arguments to rodent models, and includes a view on the current debate about reducing experimental animals overall. Please find minor comments below that might help to improve the manuscript.

For the title, I suggest to already include ‘sheep’ somewhere since a large part of the review is based on it.

The abstract needs a more stringent saying of the main points, and focus of the review; it could also already include a statements, conclusion. Please also use shorter sentences throughout the manuscript.

Please check English spelling, grammar, e.g. use ‘-‘ before (e.g. rodent-) derived, introduce abbreviations early (line 50, use ‘central’ nervous system, CNS, throughout; li 62, 89 ... 910), li 31, 58 etc. ‘in part due to’, animal model ‘for’ neurological disease, take out ‘human’ in front of patients, avert of double words, e.g. li33, effective, li182 ‘model’ etc.
